# Dynamic Distribution of Infectious Pancreatic Necrosis Virus (IPNV) Strains of Genogroups 1, 5, and 7 after Intraperitoneal Administration in Rainbow Trout (*Oncorhynchus mykiss*)

**DOI:** 10.3390/v14122634

**Published:** 2022-11-25

**Authors:** Yizhi Shao, Guangming Ren, Jingzhuang Zhao, Tongyan Lu, Qi Liu, Liming Xu

**Affiliations:** Key Laboratory of Aquatic Animal Diseases and Immune Technology of Heilongjiang Province, Department of Aquatic Animal Diseases and Control, Heilongjiang River Fisheries Research Institute, Chinese Academy of Fishery Sciences, Harbin 150070, China

**Keywords:** infectious pancreatic necrosis virus (IPNV), isolation, dynamic distribution, viral load, rainbow trout (*Oncorhynchus mykiss*)

## Abstract

Infectious pancreatic necrosis virus (IPNV) is the causative agent of rainbow trout (*Oncorhynchus mykiss*) IPN and causes significant loss of fingerlings. The currently prevalent IPNV genogroups in China are genogroups 1 and 5. However, in this study, we isolated and identified a novel IPNV, IPNV-P202019, which belonged to genogroup 7. Here, a total of 200 specific-pathogen-free rainbow trout (10 g average weight) were divided randomly into four groups to investigate the distribution of different IPNV strains (genogroups 1, 5, and 7) in 9 tissues of rainbow trout by means of intraperitoneal (ip) injection. Fish in each group were monitored after 3-, 7-, 14-, 21- and 28- days post-infection (dpi). The study showed no mortality in all groups. The distribution of IPNV genogroups 1 and 5 was similar in different tissues and had a higher number of viral loads after 3, 7, or 14 dpi. However, the distribution of IPNV genogroup 7 was detected particularly in the spleen, head kidney, and feces and had a lower number of viral loads. The results of this study provide valid data for the distribution of IPNV in rainbow trout tissues and showed that IPNV genogroups 1 and 5 were still the prevalent genogroups of IPNV in China. Although rainbow trout carried IPNV genogroup 7, the viral load was too low to be pathogenic.

## 1. Introduction

Rainbow trout (*Oncorhynchus mykiss*) is one of the most widely farmed finfish in the world and is the main species of salmonoid culture in China which plays an indispensable role in improving aquaculture industry structure, providing high-quality and high-grade aquatic products, fishery efficiency and fishermen’s income [1]. Infectious pancreatic necrosis virus (IPNV) is an economically relevant pathogen of farmed rainbow trout, which is a non-enveloped with an average size of around 60 nm, double-stranded RNA (dsRNA) virus, belongs to the family Birnaviridae and the genus aquabirnavirus [2,3]. The two dsRNA segments have been designated A and B. Segment A, with a size of about 3100 base pairs (bp), contains two open reading frames (ORFs), encodes a polyprotein of around 106 kDa composed of NH2-pVP2-VP4-VP3-COOH and a non-structural protein VP5 [4,5]. Segment B, with a size of about 2784 bp, contains a single ORF encoding the RNA-dependent RNA polymerase, VP1 [6,7]. These proteins play an important role in the process of virus invasion into host cells. First, VP2 acts as the viral attachment receptor to specifically recognize the surface cell receptor. Soon afterward, internalization is produced via macropinocytosis, and finally, viral particle transcripts and replicates inside a viral core [8].

Based on the VP2, aquabirnavirus can be divided into seven genotypes, including six genotypes [9] (genogroups 1–6) of IPNV, which are isolated from salmonid species, and one genotype (genogroup 7) [10], which is isolated from non-salmonid species comprising marine aquabirnaviruses (MABV) or yellowtail ascites virus (YTAV) [11]. The first case of IPN was reported in North America, belonging to genogroup 1 [9]. Then IPN spread all over the world. IPNV was originally isolated from a rainbow trout farm in the Shanxi province of China in the 1980s [12] and was subsequently reported by other provinces [13,14], causing significant economic losses to the rainbow trout industry in China. The main distribution of IPNV in the world is USA, genogroups 1 and 4 [9]; Canada, genogroups 1, 3, and 4 [8,15]; Mexico, genogroup 1 [16,17,18]; South America, genogroups 1 and 5 [19,20]; Europe, genogroups 1, 2, 3, 5 and 6 [3,21,22,23]; Oceania, genogroups 5 and 7 [24]; Asia, genogroups 1, 5 and 7 [9,25]. Our previous studies have shown that both IPNV genogroups 1 and 5 were found in Chinese aquaculture rainbow trout fisheries [25]. However, in 2020, during disease surveillance in rainbow trout, we found a virus with different genogroups from the above two in a fish farm in northwest China, which did not cause histopathological changes in rainbow trout. Based on preliminary laboratory tests, this is a new aquabirnavirus, which belongs to genogroup 7. In this study, this novel aquabirnavirus was isolated from these fish tissues by first-generation sequencing, and phylogenetics and the distribution of the novel aquabirnavirus and IPNV genogroups 1 and 5 in different tissues of rainbow trout were also investigated subsequently. To our knowledge, this is the first time that aquabirnavirus genogroup 7 has been isolated from cultured rainbow trout in China. At the same time, we also explored the distribution of IPNV genogroups 1, 5, and 7 in rainbow trout tissues, which increased our understanding of IPNV infection in rainbow trout.

## 2. Materials and Methods

### 2.1. Virus Isolation and Culture

Sampling was performed at a rainbow trout farm in Gansu province, China. Liver, spleen, and head kidney tissue samples were collected from rainbow trout and cut into small pieces, and homogenized in phosphate-buffered saline (PBS). The tissue homogenate was centrifuged at 4 °C at 12,000× *g* for 15 min. Then, the supernatant was filtered and sterilized with a 0.22 μm diameter filter and stored at −80 °C for later use. The Chinook salmon embryo (CHSE-214) cell line (ATCC CRL-1681) was cultured in T-25 cm^2^ cell culture flasks with medium growth medium (MEM) (GIBCO, Shanghai, China) containing 10% (*v*/*v*) fetal bovine serum (FBS) (CLARK, Shanghai, China) at 15 °C. After removing the medium from the culture flask, 85% full of monolayer cells, 1 mL of suspension was added onto the cells and incubated for 1 h at 15 °C. Then, the supernatant was removed, and 5 mL of MEM with 2% FBS was added. The infected cells were further cultured at 15 °C for a week and observed daily for the cytopathic effect (CPE). Non-infected cells were used as a control. The whole-cell culture medium was collected after freeze-thawing at −80 °C three times, followed by centrifugation at 12,000× *g* for 15 min to remove cell debris. The supernatant was collected as a viral suspension and stored for further use. IPNV-ChRtm213 (IPNV genogroup 1) [26] and IPNV-BJ2020-1 (IPNV genogroup 5) [10] have been isolated and stored in our laboratory.

### 2.2. Extraction of Viral Genome RNA

Same as in our previous study [27], virus RNA was extracted by using Trizol reagent (Invitrogen, Shanghai, China), which was obtained by the CHSE-214 cell culture. Briefly add 1 mL of Trizol reagent per suspension and incubate for 5 min. Then, add 0.2 mL chloroform, mix by shaking, and incubate for 2–3 min. Later, centrifuge for 15 min at 12,000× *g* at 4 °C and separate the aqueous phase. Add 500 μL of isopropanol to the aqueous phase and incubate for 10 min at 4 °C and centrifuge for 10 min at 12,000× *g* at 4 °C. Discard the supernatant and wash the pellet with 1 mL of 75% ethanol, then centrifuge for 5 min at 7500× *g* at 4 °C and discard the supernatant. Finally, dissolved the pellet in 50 μL of diethylpyrocarbonate (DEPC) water and stored at −80 °C until later use.

### 2.3. Virus Detection

After RNA extraction, the primers (584 bp) 5′-CAAGGCAACCGCAACYTACT-3′ (forward primer) and 5′-ATKGCAGCTGTGCACCTCAT-3′ (reverse primer) were used for IPNV detection. The one-step reverse transcription polymerase chain reaction (RT-PCR) for the detection of IPNV was performed by using PrimeScript™ One Step RT-PCR Kit Ver.2 (RR057A, TaKaRa, Dalian, China). PCR amplification was performed using Applied Biosystems (Thermo Fisher Scientific, Waltham, MA, USA). The reaction program was 94 °C 30 s, 58 °C 30 s, 72 °C 40 s, 35 cycles. RT-PCR product was analyzed by 1% agarose gel electrophoresis, and those with target band were sent to be sequenced by Sangon Biotech Shanghai Co., Ltd. (Shanghai, China).

### 2.4. Full-Length Amplification, Cloning, Nucleotide Sequencing, and Sequence Analysis

To amplify the entire IPNV genogroup 7, we designed primers and used PrimeScript™ One Step RT-PCR Kit (TaKaRa, Dalian, China). The primers are listed in Appendix A. The reaction program was 98 °C 10 s, 65 °C 10 s, 72 °C 1 min, 35 cycles. The PCR product was performed by Sangon Biotech Shanghai Co., Ltd. (Shanghai, China). Bioinformatics analysis was performed using GenBank BLAST search. The DNAMAN version 6 software (Lynnon Corporation, Quebec City, QC, Canada) and Molecular Evolutionary Genetics Analysis version 5.0 (MEGA 5.0) software were utilized to construct phylogenetic evolutionary trees.

### 2.5. Artificial Infection

In order to investigate the distribution and accumulation of IPNV of three different genogroups in rainbow trout, we conducted artificial infection. There were 200 healthy rainbow trout fingerlings, with an the average weight of 10 ± 2 g, were acclimatized to lab conditions for 2 weeks with aerated at 12 °C and fed a dry pelleted diet ad libitum before the experimental challenge. We randomly divided the fingerlings into 4 groups, with 50 individuals in each group: Group IPNV-ChRtm213: subjected to an intraperitoneally (ip) injection with IPNV-ChRtm213 (IPNV genogroup 1) (2 × 10^6^ TCID_50_ per fish, 50 μL); Group IPNV-BJ2020-1: subjected to an ip with IPNV-BJ2020-1 (IPNV genogroup 5) (2 × 10^6^ TCID_50_ per fish, 50 μL); Group IPNV-P202019: subjected to ip with IPNV-P202019 (IPNV genogroup 7) (2 × 10^6^ TCID_50_ per fish, 50 μL); Group PBS: was ip injected with phosphate buffer solution (PBS, 50 μL per fish) as a negative control. Subsequently, 6 fish were taken randomly from each group and killed on 3-, 7-, 14-, 21- and 28- days post-infection (dpi). The brain, gill, heart, liver, spleen, head kidney, feces, mucus, and muscle of each group of fish were collected and stored for further use.

### 2.6. Ethics Statement

All animal experiments were performed at the Heilongjiang Fisheries Research Institute of the Chinese Academy of Fishery Sciences in full accordance with the guidelines of the ethical review committee. According to manufacturer standard protocols, fish were anesthetized by dipping them in methane tricaine sulfonate (Sigma, Shanghai, China) prior to challenges.

### 2.7. Quantitative PCR (qPCR) Analysis of IPNV

Viral RNA was extracted in Section 2.2. The qPCR with primer pairs of VP2-F: 5′-CCACTACAGGTGGAATCAGAAC-3′, VP2-R: 5′-GATCAGTCTCCCGTAGTTGAATG-3′ and probe: FAM-GATGAGGTGCACAGCTGC-BHQ1 was used to analyze the proliferation of IPNV in rainbow trout samples using Premix Ex Taq^TM^ (Probe qPCR) (RR640A, TaKaRa, Dalian, China). Specific reaction systems and procedures refer to previous studies [1].

### 2.8. Titration

The infection titers of tissues and cell culture suspensions to be tested were determined by routine procedures on CHSE-214 monolayers [27]. The TCID_50_ of the infection samples was calculated according to the Reed and Muench method [28], the TCID_50_ of cell culture suspensions was expressed as TCID_50_/mL, and the TCID_50_ of tissue samples was expressed as TCID_50_/g.

### 2.9. Statistical Analyses

All statistical analyses were performed using SPSS 17.0 (SPSS, Chicago, IL, USA), and data are expressed as means the standard deviation (SD). One-way analysis of variance analysis (ANOVA) was performed using GraphPad Prism software (version 8.0) to determine the significance of differences between treatment groups. A *p* value of <0.05 is considered statistically significant.

## 3. Results

### 3.1. IPNV Detection in Rainbow Trout and IPNV-P202019 Isolation

IPNV was cultured in a CHSE-214 cell line from homogenates of rainbow trout tissues at 15 °C. CPE was produced at 4 dpi (Figure 1B). Non-infected cells were used as a control (Figure 1A). The whole-cell culture medium was collected, frozen, and thawed three times at −80 °C and centrifuged at 12,000× *g* for 15 min to remove cell debris. The supernatant was collected to assess the viral by using the PCR method. As shown in Figure 1C, the target gene of the PCR product obtained the expected size at 584 bp. The sequencing results were bioinformatically analyzed using GenBank blast search and showed that the virus was an aquabirnavirus which was different from the aquabirnavirus circulating in China.

### 3.2. Full-Length Genomic Sequencing and Phylogenetic Analyses

The full-length genome sequences are available on Genbank under accession No. OP272507 and OP272508. The genome is presented in its full of 3090 bp of A segment with a G + C content of 54.24% and 2738 bp of B segment with a G + C content of 52.34%. Results of multiple alignments showed that the genome sequence of the IPNV-P202019 A and B segments being studied most closely resembles YTAV Y-6 A and B segments isolated in Japan, with the pairwise nucleotide identities being 95.43% and 95.32% (Table 1 and Table 2). The genome sequence of segments A and B similarity between our virus and the IPNV-ChRtm213 isolated in China in 2013 was only 84.44% and 85.99% (Table 1 and Table 2).

Based on that, the genotypes of aquabirnavirus were determined by the VP2 gene. To further investigate the relationship between IPNV-P202019 isolated in this experiment and other aquabirnavirus family members, we performed a phylogenetic analysis of VP2 genome sequences of aquabirnavirus viruses available in GeneBank databases. Our virus was phylogenetically closely clustered with the YTAV Y-6 (AY283781) isolated in Japan, forming a sublineage of the genus Birnavirus including other marine fish aquabirnavirus isolated in Asia (Figure 2). We named this novel aquabirnavirus IPNV-P202019. In addition, we performed a phylogenetic analysis of VP1 genome sequences between IPNV-P202019 and other aquabirnavirus viruses available in GeneBank databases. Our virus was phylogenetically closely clustered with the POBV (EU161286) isolated in China. The results are shown in Appendix A.

### 3.3. Tissues Distribution of Three Genogroups of IPNV in Rainbow Trout

IPNV-ChRtm213, which titer was 10^7.1^ TCID_50_/mL, IPNV-BJ2020-1, which titer was 10^7.4^ TCID_50_/mL, IPNV-P202019, which titer was 10^7.5^ TCID_50_/mL. Were ip injected into healthy rainbow trout to determine the distribution and accumulation of IPNV of three different genogroups. No mortality was observed in either the challenge group or the control group. In order to further explore the distribution of the virus after rainbow trout infection, we collected nine different tissue samples on 3, 7, 14, 21, and 28 dpi for viral RNA and titer determination. 

#### 3.3.1. Brain

In order to determine the viral loads in the brain, samples of the brain were analyzed by qPCR (Figure 3A) and titration (Figure 3B). The results showed that rainbow trout infected with IPNV-ChRtm213 had the highest virus loads on 14 dpi compared to other time points (*p* < 0.05). Subsequently, the viral loads gradually decreased. Rainbow trout infected with IPNV-BJ2020-1 had higher virus loads on 3, 7, and 14 dpi. Later, the viral loads decreased at 21 and 28 dpi. However, rainbow trout infected with IPNV-P202019 could only be detected in the brain on 7 dpi (Figure 3A). As shown in Figure 3B, viral titers were determined by using CHSE-214 cells, and the infection titer of the detected tissue was expressed as TCID_50_/g. Concretely, the titer infected with IPNV-ChRtm213 in the brain was about 10^1.2^~10^4.1^ TCID_50_/g, 10^1.5^~10^3.0^ TCID_50_/g, which infected with IPNV-BJ2020-1 and 10^1.2^ TCID_50_/g which infected with IPNV-P202019 on 7 dpi.

#### 3.3.2. Gill

The viral loads of the gill were detected using qPCR (Figure 4A) and titration (Figure 4B). The results showed higher viral loads of IPNV-ChRtm213 on 3 and 7 dpi compared to other time points (*p* < 0.05), higher viral loads of IPNV-BJ2020-1 on 3, 7, and 14 dpi compared to 21 and 28 dpi (*p* < 0.05). However, rainbow trout infected with IPNV-P202019 could only be detected in gill on 7 dpi (Figure 4A). As shown in Figure 4B, the titer infected with IPNV-ChRtm213 in gill was about 10^2.2^~10^4.5^ TCID_50_/g, 10^1.7^~10^3.8^ TCID_50_/g which infected with IPNV-BJ2020-1 but no virus titer was detected in gill, which infected with IPNV-P202019.

#### 3.3.3. Heart

The viral loads of the heart were detected using qPCR (Figure 5A) and titration (Figure 5B). The results showed that rainbow trout infected with IPNV-ChRtm213 had higher virus loads on 3 and 7 dpi compared to other time points (*p* < 0.05). Subsequently, the viral loads gradually decreased. The virus loads of IPNV-BJ2020-1 had the highest virus loads on 7 dpi compared to other time points (*p* < 0.05). The virus loads of IPNV-P202019 had the highest virus load at 7 dpi but could not be detected after 14 dpi (Figure 5A). As shown in Figure 5B, the titer infected with IPNV-ChRtm213 in the heart was about 10^1.4^~10^4.9^ TCID_50_/g, 10^2.5^~10^6.5^ TCID_50_/g which infected with IPNV-BJ2020-1 and 10^1.3^~10^3.1^ TCID_50_/g but not after 14 dpi which infected with IPNV-P202019.

#### 3.3.4. Liver

The viral loads of the liver were detected using qPCR (Figure 6A) and titration (Figure 6B). The results showed that rainbow trout infected with IPNV-ChRtm213 had higher virus loads on 3 and 7 dpi compared to other time points (*p* < 0.05), subsequently, the viral loads gradually decreased. The viral loads of IPNV-BJ2020-1 reached a peak at 7 dpi compared to other time points (*p* < 0.05) and remained at a constant level until 28 dpi. The viral loads of IPNV-P202019 reached the peak at 7 dpi but could not detect after 14 dpi (Figure 6A). As shown in Figure 6B, the titer infected with IPNV-ChRtm213 in the liver was about 10^2.0^~10^4.7^ TCID_50_/g, 10^3.5^~10^4.6^ TCID_50_/g which infected with IPNV-BJ2020-1 and 10^1.1^~10^1.8^ TCID_50_/g but not after 14 dpi which infected with IPNV-P202019.

#### 3.3.5. Spleen

The viral loads of the spleen were detected using qPCR (Figure 7A) and titration (Figure 7B). The results showed that rainbow trout infected with IPNV-ChRtm213 had higher virus loads on 3 and 7 dpi compared to other time points (*p* < 0.05), subsequently, the viral loads gradually decreased. The virus loads of IPNV-BJ2020-1 had higher virus loads on 3, 7, and 14 dpi (*p* < 0.05), and the viral loads were still high on 21 and 28 dpi. The virus loads of IPNV-P202019 had higher virus loads on 3, 7, and 14 dpi (Figure 7A). As shown in Figure 7B, the titer infected with IPNV-ChRtm213 in the spleen was about 10^3.1^~10^5.9^ TCID_50_/g, 10^3.5^~10^6.5^ TCID_50_/g which infected with IPNV-BJ2020-1 and 10^1.2^~10^2.3^ TCID_50_/g but not after 21 dpi which infected with IPNV-P202019.

#### 3.3.6. Head Kidney

The viral loads of the head kidney were detected using qPCR (Figure 8A) and titration (Figure 8B). The results showed that rainbow trout infected with IPNV-ChRtm213 had higher virus loads on 3 dpi compared to other time points (*p* < 0.05), and the viral loads were still high on 7, 14, 21, and 28 dpi. The virus loads of IPNV-BJ2020-1 had the highest virus loads at 3, and 7 dpi, and the viral loads were still high at 14, 21, and 28 dpi. The virus loads of IPNV-P202019 had higher virus loads on 7 dpi compared to other time points (*p* < 0.05). Subsequently, the viral loads gradually decreased (Figure 8A). As shown in Figure 8B, the titer infected with IPNV-ChRtm213 in the head kidney was about 10^3.2^~10^6.1^ TCID_50_/g, 10^3.7^~10^5.6^ TCID_50_/g which infected with IPNV-BJ2020-1 and 10^1.0^~10^3.2^ TCID_50_/g but not after 28 dpi which infected with IPNV-P202019.

#### 3.3.7. Faeces

The viral loads of the feces were detected using qPCR (Figure 9A) and titration (Figure 9B). The results showed that rainbow trout infected with IPNV-ChRtm213 had higher virus loads on 3, 7, and 14 dpi compared to 21 and 28 dpi (*p* < 0.05). The virus loads of IPNV-BJ2020-1 had higher virus loads on 3 dpi compared to other time points (*p* < 0.05), and the viral loads were still high on 7, 14, 21, and 28 dpi. The virus loads of IPNV-P202019 had higher virus loads on 7 dpi compared to other time points (*p* < 0.05), subsequently, the viral loads gradually decreased (Figure 9A). As shown in Figure 9B, the titer infected with IPNV-ChRtm213 in feces was about 10^1.4^~10^3.8^ TCID_50_/g, 10^3.3^~10^4.8^ TCID_50_/g which infected with IPNV-BJ2020-1 and 10^1.2^~10^1.7^ TCID_50_/g but not after 21 dpi which infected with IPNV-P202019.

#### 3.3.8. Mucus

The viral loads of the mucus were detected using qPCR (Figure 10A) and titration (Figure 10B). The results showed that rainbow trout infected with IPNV-ChRtm213 and IPNV-BJ2020-1 both had higher virus loads on 3 and 7 dpi compared to other time points (*p* < 0.05), respectively. Rainbow trout infected with IPNV-P202019 could only be detected in mucus on 7 dpi (Figure 10A). As shown in Figure 10B, the titer infected with IPNV-ChRtm213 in mucus was about 10^1.2^~10^4.4^ TCID_50_/g, 10^2.5^~10^3.9^ TCID_50_/g which infected with IPNV-BJ2020-1 but no virus titer was detected in mucus, which infected with IPNV-P202019.

#### 3.3.9. Musculi

The viral loads of the musculi were detected using qPCR (Figure 11A) and titration (Figure 11B). The results showed that rainbow trout infected with IPNV-ChRtm213 and IPNV-BJ2020-1 both had higher virus loads on 3 and 7 dpi. Subsequently, the viral loads gradually decreased. The virus loads of IPNV-P202019 had higher virus loads on 3 dpi compared to other time points (*p* < 0.05) (Figure 11A). As shown in Figure 11B, the titer infected with IPNV-ChRtm213 in musculi was about 10^1.2^~10^4.1^ TCID_50_/g, 10^2.3^~10^3.5^ TCID_50_/g which infected with IPNV-BJ2020-1 but no virus titer was detected in musculi, which infected with IPNV-P202019.

## 4. Discussion

Numerous studies have confirmed IPN can affect salmonid aquaculture worldwide. It can result in high mortality rates of fry in first-feeding and fingerlings shortly after transfer to seawater in post-smolts [29,30]. As mentioned earlier, the pathogen of IPN, including genogroups 1–6, was called IPNV when it only infected or caused disease in salmonids [31] and called MABV or YTAV if it is isolated from non-salmonid species or isolated or caused disease in yellowtail fish (*Seriola quinqueradiata*) which was included genogroup 7 [32]. Phylogenetic analysis showed that the IPNV-P202019 strain isolated from rainbow trout in this study belonged to genogroup 7, which is not consistent with the conclusions previously obtained, indicating that aquabirnavirus genogroup 7 can be infected and isolated from salmonids, which is a new discovery.

IPNV strain can cause 80–90% mortality under natural conditions [33] but low or no mortality under artificial conditions [34]. The same results were obtained with other Chinese IPNV strains isolated from diseased fish. Further reports have documented similar phenomena [35,36,37]. Therefore, we chose high doses of IPNV (2 × 10^6^ TCID_50_) to infect rainbow trout without causing high mortality, and the above results were reasonable. 

In this study, we reported a new aquabirnavirus, IPNV-P202019. Then, we used the qPCR method and titration to detect the dynamic distribution between IPNV-P202019 (genogroup 7), IPNV-ChRtm213 (genogroup 1), and IPNV-BJ2020-1 (genogroup 5) in different tissues of rainbow trout. Our results suggested that the distribution of IPNV genogroups 1 and 5 was similar in different tissues and organs, which were detected in all tissues, and had a higher number of viral loads after 3, 7, or 14 dpi. However, the distribution of IPNV genogroup 7 was detected particularly in the spleen, head kidney, and feces and had a lower number of viral loads. Previous studies have shown that in horizontal transmission, IPNV enters through the gills, the intestinal epithelium, and/or certain areas of the skin via the surrounding water [31]. Within a few days, the virus was detected in several tissues, such as the pancreas and intestines, and proliferated rapidly in immune organs, such as the spleen and head kidney. Therefore, consistent with the above, relatively high viral loads can be detected in the head, kidney, spleen, and feces in our research. Meanwhile, the virus can also be detected in the gills, heart, and liver. The reason why IPNV is detected in most fish tissues may be that IPNV can be detected in circulating leukocytes, which can carry the virus through the blood to tissues throughout the body. However, the distribution of viruses in tissues depends largely on the fish species, the virus strains, and even the age of the fish [33]. Therefore, we can obtain the results of different viral loads in different tissues after ip injection of IPNV with three different genogroups. The results showed that IPNV genogroups 1 and 5 were still the prevalent genogroups of IPNV in China. Although the rainbow trout carried IPNV genogroup 7, the viral load was too low to be pathogenic.

## 5. Conclusions

In this study, we isolated and identified a novel aquabirnavirus from rainbow trout belonging to genogroup 7 and named IPNV-P202019. The artificial infection showed that IPNV-ChRtm213 (genogroup 1) and IPNV-BJ2020-1 (genogroup 5) could be rapidly distributed in rainbow trout tissues and had high viral loads, while IPNV-P202019 (genogroup 7) could also be detected in rainbow trout tissues, the viral loads were relatively low, indicating that IPNV genogroups 1 and 5 were still the prevalent genogroups of IPNV in China, although the rainbow trout carried IPNV genogroup 7, the viral load was too low to be pathogenic. This study provided a reference for the biological characterization, evolution, and pathogenicity of IPNV.

## Figures and Tables

**Figure 1 viruses-14-02634-f001:**
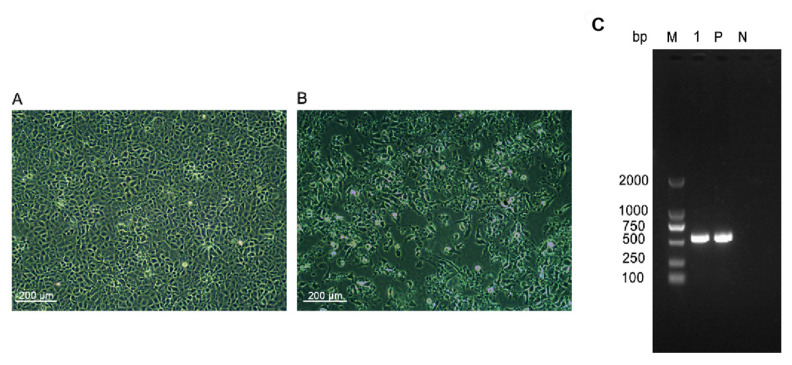
Detection of IPNV in diseased rainbow trout. (**A**) Negative control; (**B**) CPE caused by IPNV in CHSE cells were characterized at 4 d. (**C**) IPNV PCR analysis. M, DL2000 Marker; 1: sample; P, positive control; N, negative control.

**Figure 2 viruses-14-02634-f002:**
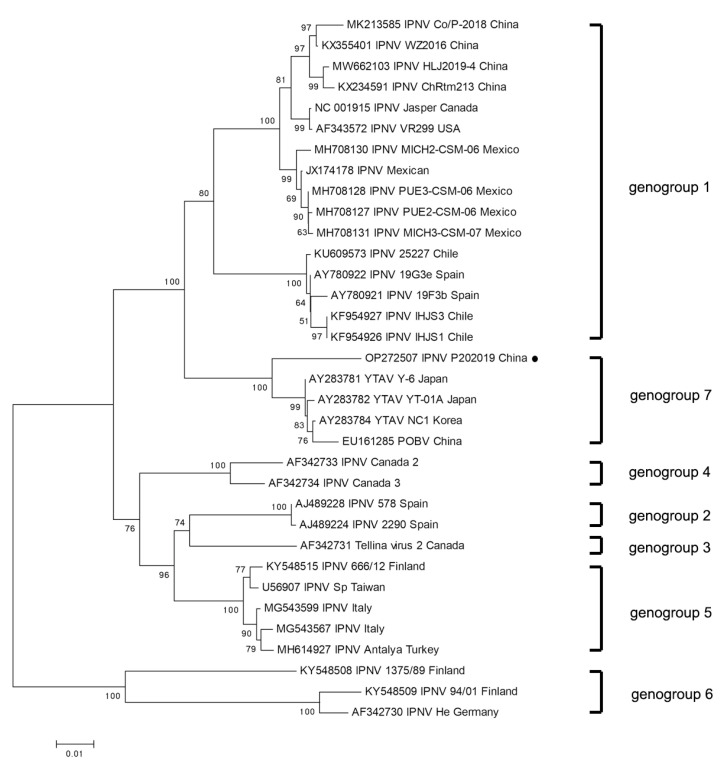
Phylogenetic analysis of IPNV-P202019 and the other representative aquabirnavirus based on nucleotide sequences of the VP2 genome. The tree was constructed by the neighbor-joining method. Bootstrap values are indicated for each node from 1000 resampling. The figure shows the name of the viruses, GenBank accession number, and country. The solid black circle represents the IPNV-P202019 isolated in this study.

**Figure 3 viruses-14-02634-f003:**
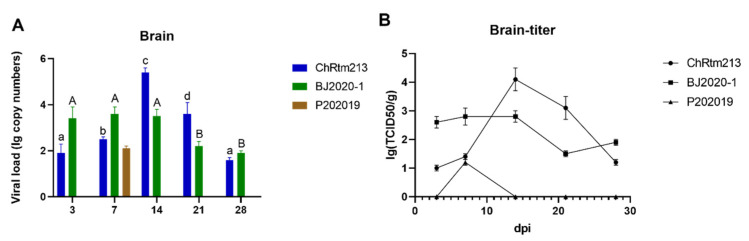
IPNV virus loads in the brain were sampled on 3, 7, 14, 21, and 28 dpi. (**A**) Replication kinetics of IPNV-infected rainbow trout. Samples were obtained in triplicate. Total RNA was extracted and analyzed by qPCR. Error bars represent the SD from triplicate within an experiment. Different letters of the same class indicated the statistical significance, *p* < 0.05. a, b, c, d indicated that the viral load of IPNV-ChRtm213 was significantly different at different sampling time points, *p* < 0.05. A, B indicated that the viral load of IPNV-BJ2020-1 was significantly different at different sampling time points, *p* < 0.05. (**B**) The detection of viral titers in the brain of IPNV-infected rainbow trout.

**Figure 4 viruses-14-02634-f004:**
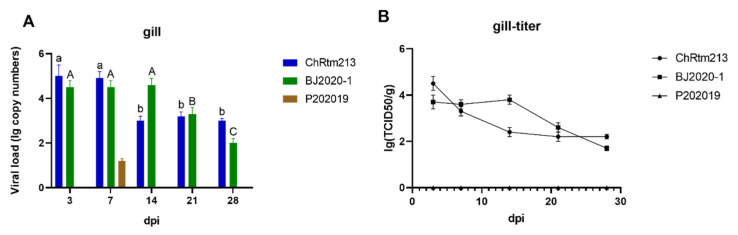
IPNV virus loads in gill sampled on 3, 7, 14, 21, and 28 dpi. (**A**) Replication kinetics of IPNV-infected rainbow trout. Samples were obtained in triplicate. Total RNA was extracted and analyzed by qPCR. Error bars represent the SD from triplicate within an experiment. Different letters of the same class indicated the statistical significance, *p* < 0.05. a, b indicated that the viral load of IPNV-ChRtm213 was significantly different at different sampling time points, *p* < 0.05. A, B, C indicated that the viral load of IPNV-BJ2020-1 was significantly different at different sampling time points, *p* < 0.05. (**B**) The detection of viral titers in the gill of IPNV-infected rainbow trout.

**Figure 5 viruses-14-02634-f005:**
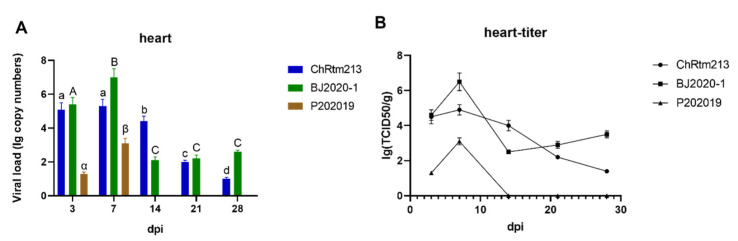
IPNV virus loads in the heart were sampled on 3, 7, 14, 21, and 28 dpi. (**A**) Replication kinetics of IPNV-infected rainbow trout. Samples were obtained in triplicate. Total RNA was extracted and analyzed by qPCR. Error bars represent the SD from triplicate within an experiment. Different letters of the same class indicated the statistical significance, *p* < 0.05. a, b, c, d indicated that the viral load of IPNV-ChRtm213 was significantly different at different sampling time points, *p* < 0.05. A, B, C indicated that the viral load of IPNV-BJ2020-1 was significantly different at different sampling time points, *p* < 0.05. α, β indicated that the viral load of IPNV-P202019 was significantly different at different sampling time points, *p* < 0.05. (**B**) The detection of viral titers in the heart of IPNV-infected rainbow trout.

**Figure 6 viruses-14-02634-f006:**
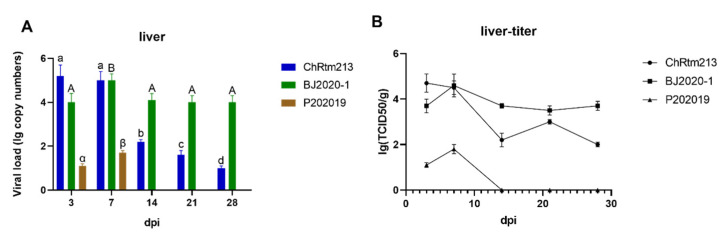
IPNV virus loads in the liver were sampled on 3, 7, 14, 21, and 28 dpi. (**A**) Replication kinetics of IPNV-infected rainbow trout. Samples were obtained in triplicate. Total RNA was extracted and analyzed by qPCR. Error bars represent the SD from triplicate within an experiment. Different letters of the same class indicated the statistical significance, *p* < 0.05. a, b, c, d indicated that the viral load of IPNV-ChRtm213 was significantly different at different sampling time points, *p* < 0.05. A, B indicated that the viral load of IPNV-BJ2020-1 was significantly different at different sampling time points, *p* < 0.05. α, β indicated that the viral load of IPNV-P202019 was significantly different at different sampling time points, *p* < 0.05. (**B**) The detection of viral titers in the liver of IPNV-infected rainbow trout.

**Figure 7 viruses-14-02634-f007:**
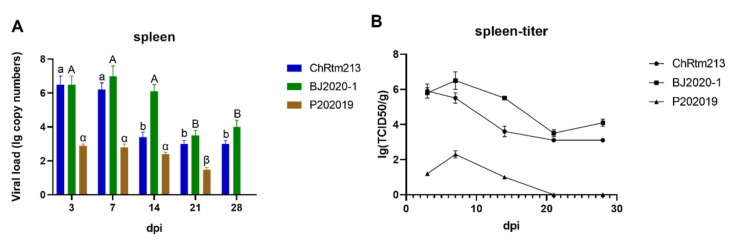
IPNV virus loads in the spleen were sampled on 3, 7, 14, 21, and 28 dpi. (**A**) Replication kinetics of IPNV-infected rainbow trout. Samples were obtained in triplicate. Total RNA was extracted and analyzed by qPCR. Error bars represent the SD from triplicate within an experiment. Different letters of the same class indicated the statistical significance, *p* < 0.05. a, b indicated that the viral load of IPNV-ChRtm213 was significantly different at different sampling time points, *p* < 0.05. A, B indicated that the viral load of IPNV-BJ2020-1 was significantly different at different sampling time points, *p* < 0.05. α, β indicated that the viral load of IPNV-P202019 was significantly different at different sampling time points, *p* < 0.05. (**B**) The detection of viral titers in the spleen of IPNV-infected rainbow trout.

**Figure 8 viruses-14-02634-f008:**
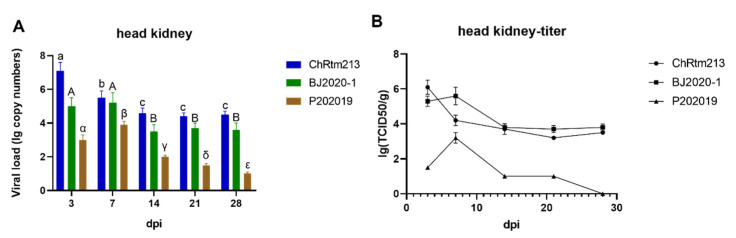
IPNV virus loads in the head kidney were sampled on 3, 7, 14, 21, and 28 dpi. (**A**) Replication kinetics of IPNV-infected rainbow trout. Samples were obtained in triplicate. Total RNA was extracted and analyzed by qPCR. Error bars represent the SD from triplicate within an experiment. Different letters of the same class indicated the statistical significance, *p* < 0.05. a, b, c indicated that the viral load of IPNV-ChRtm213 was significantly different at different sampling time points, *p* < 0.05. A, B indicated that the viral load of IPNV-BJ2020-1 was significantly different at different sampling time points, *p* < 0.05. α, β, γ, δ, ε indicated that the viral load of IPNV-P202019 was significantly different at different sampling time points, *p* < 0.05. (**B**) The detection of viral titers in the head kidney of IPNV-infected rainbow trout.

**Figure 9 viruses-14-02634-f009:**
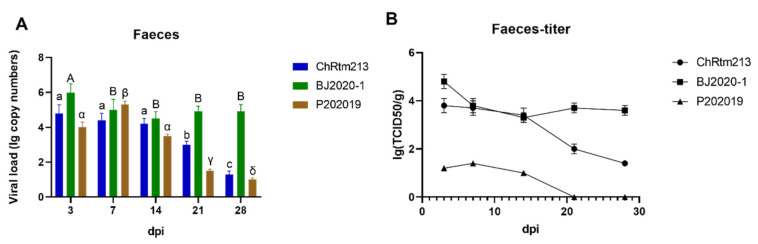
IPNV virus loads in feces were sampled on 3, 7, 14, 21, and 28 dpi. (**A**) Replication kinetics of IPNV-infected rainbow trout. Samples were obtained in triplicate. Total RNA was extracted and analyzed by qPCR. Error bars represent the SD from triplicate within an experiment. Different letters of the same class indicated the statistical significance, *p* < 0.05. a, b, c indicated that the viral load of IPNV-ChRtm213 was significantly different at different sampling time points, *p* < 0.05. A, B indicated that the viral load of IPNV-BJ2020-1 was significantly different at different sampling time points, *p* < 0.05. α, β, γ, δ indicated that the viral load of IPNV-P202019 was significantly different at different sampling time points, *p* < 0.05. (**B**) The detection of viral titers in feces of IPNV-infected rainbow trout.

**Figure 10 viruses-14-02634-f010:**
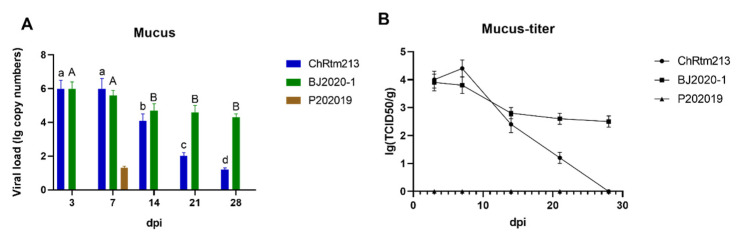
IPNV virus loads in mucus were sampled on 3, 7, 14, 21, and 28 dpi. (**A**) Replication kinetics of IPNV-infected rainbow trout. Samples were obtained in triplicate. Total RNA was extracted and analyzed by qPCR. Error bars represent the SD from triplicate within an experiment. Different letters of the same class indicated the statistical significance, *p* < 0.05. a, b, c, d indicated that the viral load of IPNV-ChRtm213 was significantly different at different sampling time points, *p* < 0.05. A, B indicated that the viral load of IPNV-BJ2020-1 was significantly different at different sampling time points, *p* < 0.05. (**B**) The detection of viral titers in the mucus of IPNV-infected rainbow trout.

**Figure 11 viruses-14-02634-f011:**
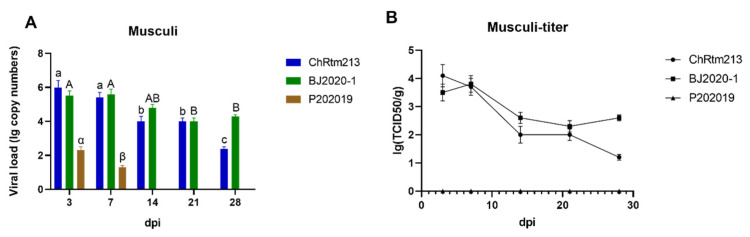
IPNV virus loads in musculi were sampled on 3, 7, 14, 21, and 28 dpi. (**A**) Replication kinetics of IPNV-infected rainbow trout. Samples were obtained in triplicate. Total RNA was extracted and analyzed by qPCR. Error bars represent the SD from triplicate within an experiment. Different letters of the same class indicated the statistical significance, *p* < 0.05. a, b, c indicated that the viral load of IPNV-ChRtm213 was significantly different at different sampling time points, *p* < 0.05. A, B indicated that the viral load of IPNV-BJ2020-1 was significantly different at different sampling time points, *p* < 0.05. α, β indicated that the viral load of IPNV-P202019 was significantly different at different sampling time points, *p* < 0.05. (**B**) The detection of viral titers in musculi of IPNV-infected rainbow trout.

**Table 1 viruses-14-02634-t001:** Nucleotide identities for the full-length genome of IPNV-P202019 A segment.

Strain	Genotype	Similarity with P202019 A Segment (%)	GenBank Accession No.	G + C (%)
IPNV-P202019	7	100	OP272507	54.24
YTAV Y-6	7	95.43	AY283781	54.42
MABV H1	7	95.40	AY283783	54.52
POBV	7	94.56	EU161285	54.00
IPNV ChRtm213	1	84.44	KX234591	55.02
IPNV AM-98	1	84.15	AY283780	55.30
IPNV PA1	1	83.05	MH010544	54.90
IPNV VR299	1	83.94	AF343572	54.92
IPNV Antalya	5	79.54	MH614927	55.65
MEIPNV1310	5	79.00	KY315690	55.69
S-IPNV/FS12-01	5	78.48	DQ536090	55.61
IPNV Connecticut-1	4	76.28	JF440810	54.36

**Table 2 viruses-14-02634-t002:** Nucleotide identities for the full-length genome of IPNV-P202019 B segment.

Strain	Genotype	Similarity with P202019 B Segment (%)	GenBank Accession No.	G + C (%)
IPNV-P202019	7	100	OP272508	52.34
YTAV Y-6	7	95.32	AY129662	52.65
YTAV H1	7	95.25	AY129665	52.76
YTAV NC1	7	94.99	AY129666	52.32
IPNV ChRtm213	1	85.99	KX234590	53.60
IPNV 17F2	1	85.54	AY780927	54.22
IPNV PA1	1	84.35	MH010545	54.10
IPNV 20G1d	1	80.65	AY780931	53.41
IPNV_SWE16_GG6	6	76.96	ON409688	51.68

## Data Availability

Not applicable.

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
