# Peer review of "Dynamic Distribution of Infectious Pancreatic Necrosis Virus (IPNV) Strains of Genogroups 1, 5, and 7 after Intraperitoneal Administration in Rainbow Trout (*Oncorhynchus mykiss*)"

_viruses, 2022, doi:10.3390/v14122634_

Round 1

Reviewer 1 Report

  In this study, authors studied "Dynamic distribution of infectious pancreatic necrosis virus 2 (IPNV) strains of genogroups 1, 5 and 7 after intraperitoneal ad- 3 ministration in rainbow trout (Oncorhynchus mykiss)". Based on this field, other data should be further added before it can be published.

1.     No Scale in Fig. 1

2.     Since authors identified the novel virus, the symptom of fish should be described by photos.

3.     The histopathology pictures of the tissues of the viral-infected fish should be added.

4.     The characters of virus (e.g., virus-induced apoptosis) should be studied.

5.     The viral particle can invade in the nucleus or cytoplasm?

6.     The infectious numbers of the virus obtained here in cells? Multiple methods should be used, such as Western blot, Single-Particle Imaging Assay.

Reviewer 2 Report

In this study, the author isolated IPNV genotype 7 from rainbow trout for the first time, and studied the dynamic distribution of the existing 1, 5 and 7 genotypes of IPNV in rainbow trout, which has practical significance. Before the article can be accepted, the following points need to be revised in the article. And the manuscript should be revised by a native professional English speaker before re-submission.

1. Line 62-71,  the sentences about the results are suggested to be removed from the introduction section.

2. Line 92,  if no reference added here, the words Same as our previous study is suggested deleted. 

3.Figure 2, the isolated virus was sequenced and phylogenetic analysis is based on VP2, while it is advisable to perform it also for VP1, in order to exclude reassortment events and give a better characterization of the new IPN.

4. In materials and methods, the author mentioned that statistical analysis was made on the experimental results of this research. However, in results, the author only marked in the figure without detailed text description, please make appropriate supplement.

5.In line 392-398, these sentences are highly similarity with preface. Please check and rewrite the sentences.
